RNase H-dependent amplification improves the accuracy of rolling circle amplification combined with loop-mediated isothermal amplification (RCA-LAMP)

Hasegawa Takema h5takema@gmail.com 1 3
Hapsari Diana 1
Iwahashi Hitoshi 2
1 The United Graduate School of Agricultural Science, Gifu University , Gifu , Gifu , Japan
2 Faculty of Applied Biological Sciences, Gifu University , Gifu , Gifu , Japan
3 Current affiliation: Research Institute for Material and Chemical Measurement, National Metrology Institute of Japan (NMIJ), National Institute of Advanced Industrial Science and Technology (AIST) , Tsukuba , Ibaraki , Japan
Sotelo-Mundo Rogerio
Electronic publication date: 2021 Jul 30
Publication date: 2021
Volume: 9
Electronic Location ID: e11851
Received 2021 Mar 18; Accepted 2021 Jul 3
Copyright: ©2021 Hasegawa et al.
Copyright year: 2021
Copyright holder: Hasegawa et al.
License: This is an open access article distributed under the terms of the Creative Commons Attribution License, which permits unrestricted use, distribution, reproduction and adaptation in any medium and for any purpose provided that it is properly attributed. For attribution, the original author(s), title, publication source (PeerJ) and either DOI or URL of the article must be cited.
License URL: https://creativecommons.org/licenses/by/4.0/

Keywords: RNA measurement, Rolling circle amplification, Loop-mediated isothermal amplification, RNase H-dependent PCR, Splint R ligase

Funding: The authors received no funding for this work.

==============================
The hybrid method upon combining rolling circle amplification and loop-mediated isothermal amplification (RCA-LAMP) was developed to quantify very small amount of different type of RNAs, such as miRNAs. RCA-LAMP can help detect short sequences through padlock probe (PLP) circularization and exhibit powerful DNA amplification. However, one of the factors that determines the detection limit of RCA-LAMP is non-specific amplification. In this study, we improved the accuracy of RCA-LAMP through applying RNase H-dependent PCR (rhPCR) technology. In this method, the non-specific amplification was suppressed by using the rh primer, which is designed through blocking the modification at the 3′end to stop DNA polymerase reaction and replacing the 6th DNA molecule from the end with RNA using RNase H2 enzyme. Traditional RCA-LAMP amplified the non-specific amplicons from linear PLP without a targeting reaction, while RCA-LAMP with rh primer and RNase H2 suppressed the non-specific amplification. Conversely, we identified the risk posed upon conducting PLP cyclization reaction using Splint R ligase in the RNA-targeting step that occurred even in the RNA-negative condition, which is another factor determining the detection limit of RCA-LAMP. Therefore, this study contributes in improving the accuracy of RNA quantification using RCA-LAMP.

Introduction

Isothermal amplification methods have been studied and applied to quantify RNA (Yan et al., 2014). For example, reverse transcription loop-mediated isothermal amplification (RT-LAMP) has been used for the detection of RNA viruses (Fukuta et al., 2003; Curtis, Rudolph & Owen, 2008). LAMP amplifies DNA from the rolling circle amplification (RCA), which is used for miRNA and mRNA detection (Jonstrup, Koch & Kjems, 2006; Cheng et al., 2009; Christian et al., 2001; Deng et al., 2017). This method’s characteristic feature that the DNA template has a loop structure at both ends (Notomi et al., 2000). RCA amplifies long-chain single-strand DNA from a circular single-stranded DNA template (Lizardi et al., 1998). One of the advantages of isothermal amplification is the fast DNA amplification speed, which is because an optimum temperature is maintained throughout the reaction (Karami et al., 2011).

Recently, RCA-LAMP hybrid method has been extensively studied and applied (Ruff et al., 2006; Tian et al., 2019). Figure 1 shows the mechanism of RCA-LAMP. First padlock probe PLP , a single-strand DNA probe designed with complementary sequences at both ends, hybridizes to the target RNA, and then, both ends of PLP are ligated using ligase to circularize. Second, the strand-displacement DNA polymerase synthesizes long-chain single-strand DNA using circular PLP as a template. This long-chain single-stranded DNA exhibits a loop structure at the 5′  end by using a specifically designed primer to form a loop structure. Third, many LAMP DNA templates are synthesized from long-chain single-strand DNA. LAMP reaction occurs for each LAMP DNA template.

Figure 1 Schematic illustration of the RCA-LAMP applied rhPCR.

The triangles indicate the PLP junction.

The advantages of RCA-LAMP are that it can detect very short RNA molecules, such as miRNAs, and exhibits high amplification power. Conversely, the major limitation of RCA-LAMP is the low accuracy of amplification, as there is a risk of synthesizing the LAMP DNA template through non-specific DNA synthesis using linear PLP and primers. In previous studies, DNA amplification was confirmed in RNA-negative samples, used to determine the detection limit of RNA (Tian et al., 2019). To improve the sensitivity of RNA quantification using this method, it is important to improve RCA-LAMP accuracy. Therefore, in this study, we aimed to improve the accuracy of RCA-LAMP.

In this study, we used Splint R ligase to circularize PLP and target RNA. Splint R binds a DNA nick on the DNA-RNA complementary strand (Lohman et al., 2014). Some reports suggest that Splint R is more accurate than the traditional ligase when applied in RCA (Deng et al., 2017; Jin et al., 2016; Takahashi et al., 2018). Moreover, we used RNA Solutions by Qualitative Analysis (AIST, Japan) as the model target RNA. This standard RNA is designed based on human mRNA and exhibits low homology with natural sequences (Tong et al., 2006).

For this, we focused on the technique of RNase H-dependent PCR (rhPCR) (Dobosy et al., 2001). In this method, rh primer, which is designed to block modification at the 3′end to stop DNA polymerase reaction and replace the 6th DNA molecule from the end with RNA using RNase H2 enzyme is used. RNase H2 recognizes a perfect complementary double-strand and cleaves RNA precisely where the rh primer binds. DNA polymerase can then synthesize DNA by removing the blocking modification. Therefore, rhPCR helps prevent non-specific amplification. For this reason, to improve the accuracy of RCA-LAMP, we applied rhPCR (Fig. 1).

In this study, we first discuss the reason why non-specific amplification occurs in a traditional RCA-LAMP assay. Furthermore, we show that through applying the rhPCR technique, the accuracy of RCA-LAMP is improved.

Materials & Methods

Materials

Splint R ligase, Bst 2.0 polymerase, and dNTP mix were purchased from New England Biolabs (NEB, Ipswich, MA, USA). RNase H2, from Integrated DNA Technologies (Iowa, USA), EvaGreen fluorescent dye from CosmoBio (Tokyo, Japan), and betaine from Fijufilm Wako Chemical (Osaka, Japan) were also purchased.

Design of DNA and RNA sequences

RNA Solutions by Qualitative Analysis was purchased from AIST (Ibaraki, Japan) (hereafter known as “standard RNA”) to be used as target RNA. This standard RNA is developed Certified reference material (CRM) (National Metrology Institute of Japan (NMIJ), Japan 2013). This standard RNA is available in five different types, including 500-A, 500-B, and 500-C that are 533 nt each and 1000-A, and 1000-B that are 1033 nt each. In this study, we used 1000-B. Table 1 shows PLP and primer sequences. PLP was synthesized and purified using Fasmac (Kanagawa, Japan). Primers were synthesized and purified by Integrated DNA Technologies ((Redwood City, CA, USA).

Table 1 Padlock probe and primers sequence.

Underlines are homologous to the ribonucleotide sequence. rA and rG are RNA. X is C3 spacer.

	Sequence	
Padlock probe	phos-GCATCGAACATTTTTGGAACTCTGCTCGACAAACGAC
ACGACACGACATTTCCCTAACCCTAACCCATTTGTCTGCCC
ACAACCTTTCTCTTACGAATC	
Primer set A Fw	CCCTAACCCATTTGTCTGCTTTGTTTGTCGAGCAGAGTTCC	
Primer set A Rv	GGGAAATGTCGTGTCGTGAAACACAACCTTTCTCTTACGAATC	
Primer set B Fw	CACGACATTTCCCTAACCCAAAGTTCCAAAAATGTTCGATG	
Primer set B Rv	GTGTCGTTTGTCGAGCAGTTTATTTGTCTGCCCACAACC	
rh primer Fw	CCCTAACCCATTTGTCTGCTTTGTTTGTCGAGCAGAGTTCCrAAAAAC-X	
rh primer Rv	GGGAAATGTCGTGTCGTGAAACACAACCTTTCTCTTACGAATCrGCATCC-X	

DNA amplification using RCA-LAMP

For PLP-mediated targeting, 1 µL of 10x reaction buffer, 2 µL of 10 nM PLP, and 5 µL of RNA were mixed. The mixture was incubated at 90 °C for 1 min, 70 °C for 1 min, and cooled to room temperature for 45 min. After that, 2 µL of Splint R ligase (6.75 U/µL) was added and incubated at 37 °C for 30 min for ligation, inactivated at 65 °C for 20 min and cooled at 4 °C.

For RCA-LAMP amplification, 1x reaction buffer, 1 mM each dNTP, 6 mM MgSO4, 0.8 M betaine, 0.8 µM forward and reverse primer, 4U Bst 2.0 polymerase, and 1 µL of PLP ligation products were mixed and incubated at 68 °C for 3 h. The primers were replaced with rh primer, and 50 mU RNase H2 enzyme was added to run the rhPCR reaction for RCA-LAMP. Then, the DNA amplicons were analyzed using 2% agarose gel electrophoresis and stained with ethidium bromide.

Quick-Load Purple 1 kb Plus DNA Ladder (0.1. 10.0 kb) purchased from New England Biolabs (NEB; Massachusetts, USA) was used as the DNA ladder marker. Also, 1x EvaGreen was added to quantify DNA amplification in real-time. The fluorescence intensity of the RCA-LAMP reaction system was monitored in real-time using the Applied Biosystems Step One Plus real-time PCR system (Thermo Fisher Scientific) for 3 h at intervals of 1 min.

DNA sequencing

DNA amplicons were purified using the Fastgene gel/PCR extraction kit purchased from Nippon Genetics (Tokyo, Japan). The purified amplicons were provided to the Division of Genomics Research of Gifu University, which runs the DNA Sequencer ABI 3130 to read the sequences.

Results

Amplification of non-specific amplicon in RCA-LAMP

In this study, we used the standard RNA, designed as a poly (A) tail forming an mRNA, as the target RNA. Splint R ligase was used for the circularization of PLP. RCA-LAMP system was designed using only a few primers. Two types of primer sets were designed to elucidate the causes of non-specific amplification (Figs. 2A, 2B). In primer set A, the forward primer targeted the middle portion of PLP, and the reverse primer targeted the sequence of RNA-targeting region. In primer set B, the forward primer targeted the RNA-target region sequence and the reverse primer targeted the middle portion of PLP.

Figure 2 DNA amplification using traditional RCA-LAMP and the sequence of non-specific amplicon.

(A) Design of the primer set A. (B) Design of the primer set B. Small letter c: complimentary sequence; Fw: Forward primer; Rv: Reverse primer; T5′and T3′: RNA targeting region. (C) DNA amplicon generated using RCA-LAMP with primer sets A and B. M: DNA ladder marker. (D) Sequence of amplicon generated using linear PLP with primer set A. (E) Sequence of amplicon generated using linear PLP with primer set B. (F) The sequence of PLP and forward primer of the primer set B.

To elucidate the cause of non-specific amplification, we read a sequence of non-specific amplicons. DNA amplification using RCA-LAMP should occur from circular PLP rather than from linear PLP. However, DNA amplification was observed from the linear PLP for both the primer sets (Fig. 2C). To develop a more sensitive RNA detection method using RCA-LAMP, it is crucial to suppress non-specific amplification. Figures 2C and 2D show the resulting sequence of the nonspecific amplicon. The sequence of non-specific amplicons generated using primer set A was random (Fig. 2D), which means that the polymerase reaction started at a random location. It is speculated that in this reaction the LAMP reaction template, which has two stem-loops at both ends, was synthesized. Conversely, the sequence of non-specific amplicons generated using the primer set B exhibited clear peaks (Fig. 2E). Also, it was replaced with the primer sequence at an unexpected position, that is from the middle of the PLP sequence, which means that DNA polymerase synthesized DNA from the 3′  end of the primer using PLP as a template. The 4th to 7th base sequences on the 3′  end of the forward primer had complementary sequences at the site where non-specific amplification occurred on the PLP (Fig. 2F). However, the 3′  end of the forward primer was not complementary. Thus, it was speculated that the annealing of the four bases present in the middle of the primer onto PLP increased the risk of non-specific amplification. These results indicate that it is difficult to improve the specificity of the reaction only by changing the sequence of the primers.

rhPCR technique improves the specificity of RCA-LAMP

To suppress non-specific amplification, we utilized rhPCR technique. For this, RNase H2 enzyme and rh primer with blocking modification at the 3′  end and 6th DNA from the 3′  end replaced with RNA are used. RNase H2 cleaves the replaced RNA only when its periphery forms a complete complementary strand. The blocking modification is released from the primer when the RNA is cleaved, following the amplification reaction stars. The DNA amplification reaction initiates only when the primer is accurately annealed to the target region. Therefore, rhPCR was expected to suppress the initiation of non-specific amplification in RCA-LAMP and stop the subsequent amplification reaction even if non-specific amplification occurs.

Since the junction region sequence of PLP gets excised when Splint R ligase is used, the specificity can be improved by designing the rh primer’s target region targeting the junction region of PLP. The rh primer should work as a primer for DNA template generated from circular PLP. If non-specific amplification occurs, the rh primer should stop the amplification reaction. The reverse primer of primer set A was designed using the rh primer because the target region was designed to target the junction region of PLP (Fig. 3A). The forward primer of primer set A is not targeted to the junction region of PLP. However, the rh primer has the potential to stop the non-specific amplification (Fig. 3B).

Figure 3 rhPCR technique improves specificity of RCA-LAMP.

(A) Primer designed with rh primer to replace reverse primer was called rh (+,-). (B) Primer designed with rh primer to replace both primers was called rh (+,+). (C) Amplicon generated using RCA-LAMP with rh primer. M indicates the DNA ladder marker. C indicates the sample in which circular PLP was used as a template DNA. L indicates the sample in which linear PLP was used as a template DNA.

Therefore, the reaction systems in which the reverse primer was replaced with rh primer and both the primers were replaced with rh primer were applied to conduct RCA-LAMP. Simultaneously, the forward primer of primer set B was designed to target the junction region of PLP. Replacing this primer with the rh primer will make the 3′  end of the rh primer complementary to the four bases of the 3′  end of PLP. This complementary sequence of both 3′  ends could lead to DNA polymerase reaction initiation from the 3′  end of PLP. The newly synthesized double-stranded DNA provides a target for RNase H2, and blocking of the modification of rh primer is released. Therefore, primer set B was not replaced with rh primer.

Figure 3C shows the DNA amplification from each PLP using RCA-LAMP with rh primer. Traditional RCA-LAMP was used to amplify DNA from linear PLP. RCA-LAMP with rh primer did not amplify DNA from linear PLP (reaction time was 3 h), and showed that rh primer suppressed the non-specific amplification of the linear PLP. Therefore, this result suggests that the rh primer contributed in improving the amplification accuracy of RCA-LAMP.

Real-time quantification using RCA-LAMP with rh primer

The analytical performance of RCA-LAMP with rh primer was evaluated by detecting standard RNA at different concentrations. The samples with Splint R ligase reaction without RNA and Splint R ligase and RNA (linear PLP) were also evaluated. Figs. 4A–4C shows that the real-time fluorescence curves upon DNA amplification changed with standard RNA concentrations in the range of 1 nM to 100 fM. DNA amplification was in proportion to the concentration of target RNA. Non-specific amplification from linear PLP (Splint R (-), RNA (-)) was suppressed using the rh primer. DNA was amplified from PLP, which reacted with Splint R ligase without RNA sample (RNA (-)). This result suggests that Splint R ligated the ends of DNA without RNA.

Figure 4 Real-time quantification using RCA-LAMP with rh primer.

(A) RCA-LAMP with rh(+, -). (B) RCA-LAMP with rh(+,+). (C) RCA-LAMP with rh(-,-). (D) Ct value of each primer set. rh(+,+) and rh(+,-) had no value in the RNA(-) ligase (-) sample because there was no amplification (n = 3).

Figure 4D shows the relationship between Ct value and target RNA concentration. The amplification speed of the traditional RCA-LAMP using rh (-, -) was the fastest, and using rh (+,+) was the slowest. The coefficient of determination (R2) of the calibration curve within 1 nM to 1 pM of RCA-LAMP using rh (+, -), rh (+,+), and rh (-,-) were 0.999, 0.977, and 0.732, respectively. RCA-LAMP using rh (+, -) was the most accurate. The limit of detection of RCA-LAMP using rh (+, -) was 1 pM. Non-specific amplicons from linear PLP (Splint R (-), RNA (-)) with traditional RCA-LAMP using rh(-,-) were amplified at various speeds (Fig. 4D). The rh primer could accurately suppress the non-specific amplification.

Discussion

In this study, we used the technique of rhPCR to improve the accuracy of RCA-LAMP. We showed that rh primer and RNase H2 enzyme suppressed the non-specific amplification of linear PLP.

RCA-LAMP using rh primer undergoes two kinds of enzyme reactions. First, the cleavage of the RNA portion complementary to the rh primer through RNase H2. Second, DNA amplification through strand-displacement DNA polymerase (here we used Bst DNA polymerase). It is considered that the speed of DNA amplification using rh (+,+) primer set is slower than that using rh (+,-) as the amplification mediated by rh (+,+) requires more RNase H2 enzyme than rh (+,-). Therefore, the reaction may be speedier if the amount of RNase H2 is increased; however it would make the cost high, and also, the accuracy needs to be examined.

The calibration curve of amplification using rh (+, -) was more accurate than that using rh (+,+). The reason is that the amplification using rh (+,-) exhibits a more direct reaction system than that using rh (+,+). Since the forward primer used is a regular primer, RCA reaction is quick. Amplification using rh (+,-) could only generate an accurate calibration curve up to 1 pM. For this, the following two reasons are projected. One is that the reaction of RCA-LAMP is too robust, and thus, it poses the risk of amplifying non-specific sequences. The other is that the reaction of RCA-LAMP with rh primer is complex. For instance, it is not easy to estimate the amount of RNA and the DNA amplification rate in a reaction with many steps. These could be the attributing reasons that the amplification using rh(+,-) exhibits a more accurate calibration curve than that using rh(+,+). Therefore, it is considered that the traditional RCA-LAMP could not produce an accurate calibration curve due to non-specific amplification.

Splint R ligase was used to circularize PLP. However, DNA was amplified from PLP in a reaction with Splint R to RNA. Conversely, DNA was not amplified from PLP in a reaction without Splint R and RNA. This result indicates that Splint R poses a risk of executing the reaction without the presence of RNA. It has previously been reported that Splint R is more sensitive than other DNA ligases as it can detect RNA directly (Deng et al., 2017; Jin et al., 2016; Takahashi et al., 2018). Therefore, it is important to study how Splint R reaction does not react negatively with RNA to improve the accuracy.

A calibration curve could not be generated using rhPCR for small amounts of RNA; however, the difference between circular and linear PLP could be assessed using RCA-LAMP. Thus, it can be applied for digital quantification of RNA because digital quantification can disregard the DNA amplification speed and consider only positive or negative amplification (Vogelstein & Kinzler, 1999; Hindson et al., 2011; Quan, Sauzade & Brouzes, 2018). RCA-LAMP combined with rh primer results in robust DNA amplification and can determine PLP cyclization associated with digital quantification. For this, we need to improve the accuracy of targeting RNA using PLP.

Conclusions

The technique of rhPCR suppressed the non-specific amplification from linear PLP in RCA-LAMP. It contributes to the improving signal-to-noise ration of RNA quantification by RCA-LAMP.

We would like to thank Editage for English language editing.

Additional Information and Declarations

Competing Interests

Author Contributions

DNA Deposition

Data Availability

The authors declare there are no competing interests.

Takema Hasegawa conceived and designed the experiments, performed the experiments, analyzed the data, prepared figures and/or tables, and approved the final draft.

Diana Hapsari and Hitoshi Iwahashi conceived and designed the experiments, authored or reviewed drafts of the paper, and approved the final draft.

The following information was supplied regarding the deposition of DNA sequences:

The standard RNA 1000B sequence is available at GenBank: AB610947.1.

The following information was supplied regarding data availability:

All data are included in this article.

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
