# Peer review of "RNase H-dependent amplification improves the accuracy of rolling circle amplification combined with loop-mediated isothermal amplification (RCA-LAMP)"

_PeerJ, doi:10.7717/peerj.11851_

## Round 0.1 · original submission · Major Revisions

Please take into consideration the reviewer’s comments and provide back a point-by-point rebuttal letter addressing those concerns.

·

Basic reporting

The manuscript has a clear and professional use of English; some suggestions to improve some sentences are given in the word file. The references used in the manuscript are the adequate to show how accuracy of a method can be obtained. The paper´s structure is good, and figures are clear and concise.
In line 189 of the word file, it is unclear who or what "This" refers to, could you please consider rewriting the sentence to make it clearer?

Experimental design

The methods are clearly described, the DNA and RNA sequences designed and used are clearly presented. Conditions for the amplification reactions are detailed and clearly presented.

Validity of the findings

An improved RCA-LAMP technique with improved accuracy is presented in the manuscript with the introduction of RNase H-dependent PCR. This technology is desirable specially when amplifying very small amount of RNAs, like small RNAs.

I consider this improvement important, since non-specific amplification is not desirable in PCR reactions. I think the data presented is robust and conclusion is clearly stated and supported by the data.

Additional comments

The manuscript presents in a clear way how to improve the accuracy of the RCA-LAMP isothermal amplification to quantify RNA. For the real-time quantification did you did any statistics? I did not see at the manuscript the number of technical replicates for the experiments.

Reviewer 2 ·

Basic reporting

no comment

Experimental design

no comment

Validity of the findings

no comment

Additional comments

The authors present an interesting and straightforward set of experiments. They focused on the technique of RNase H-dependent PCR (rhPCR). By applying this technique, the accuracy of RCA-LAMP is improved. They found rh primer and RNase H2 enzyme suppressed the non-specific amplification of linear PLP. Further, non-specific amplification in a traditional RCA-LAMP assay was discussed. As such, the work presented in this manuscript provide useful information for researchers in this field.

Reviewer 3 ·

Basic reporting

In this manuscript, the authors proposed the method for suppressing the non-specific amplification signal in RCA-LAMP technology, which relies on RNase H-dependent PCR primer reported in 2011 (ref 1). The proposed idea is worthy of publication, but there are many missing and incorrect results in the manuscript. Therefore, I suggest the acceptance of this manuscript after accommodating the following comments.

Ref 1) BMC Biotechnology, 2011, 11: 80

Experimental design

1) The title should be changed because it sounds like PCR is used in this work.
2) In Figure 3 & 4, the authors did not test the forward primer with RNA modification only (rh (-,+)). The authors need to provide the results obtained by RCA-LAMP with rh (-,+).
3) The authors added RNA at the 6th from the 3’ end of primer. In the original rHPCR paper (ref 1), RNA is added at the 5th from the 3’ end of primer. The authors need to provide the reason for this or optimization results.

Validity of the findings

1) In Figure 4, the authors need to provide more details about how to calculate POI
2) In Figure 4(D), xrh (-,+) should be corrected into xrh (+,-). In addition, Figure 4(A-C) do not match with the results in Figure 4(D), and the line color in Figure 4(A-C) is unclear. The authors need to change these mistakes.
3) The authors need to calculate the limit of detection, and test the target RNA at lower concentrations (e.g. 10 fM, 1 fM, 100 aM). In the original RCA-LAMP paper (ref 2; Biosensors and Bioelectronics, 2019, 128: 17-22) achieved the limit of detection of 10 aM without rHPCR technique, which seems better than the one in this work.
4) The authors emphasized that the use of RNA incorporated primer can suppress the non-specific amplification signal produced in the absence of target RNA, but undesirable amplification in the absence of RNA was observed in Figure 4. The authors need to discuss why these background signals do not completely disappear.
5) The authors need to check the specificity of the proposed system by using the control RNA samples.
6) Instead of listing the primer sequences, the authors need to make the table for clarity.

Additional comments

In this manuscript, the authors proposed the method for suppressing the non-specific amplification signal in RCA-LAMP technology, which relies on RNase H-dependent PCR primer reported in 2011 (ref 1). The proposed idea is worthy of publication, but there are many missing and incorrect results in the manuscript. Therefore, I suggest the acceptance of this manuscript after accommodating the following comments.

Ref 1) BMC Biotechnology, 2011, 11: 80
Ref 2) Biosensors and Bioelectronics, 2019, 128: 17-22

1) The title should be changed because it sounds like PCR is used in this work.
2) In Figure 3 & 4, the authors did not test the forward primer with RNA modification only (rh (-,+)). The authors need to provide the results obtained by RCA-LAMP with rh (-,+).
3) The authors added RNA at the 6th from the 3’ end of primer. In the original rHPCR paper (ref 1), RNA is added at the 5th from the 3’ end of primer. The authors need to provide the reason for this or optimization results.
4) In Figure 4, the authors need to provide more details about how to calculate POI
5) In Figure 4(D), xrh (-,+) should be corrected into xrh (+,-). In addition, Figure 4(A-C) do not match with the results in Figure 4(D), and the line color in Figure 4(A-C) is unclear. The authors need to change these mistakes.
6) The authors need to calculate the limit of detection, and test the target RNA at lower concentrations (e.g. 10 fM, 1 fM, 100 aM). In the original RCA-LAMP paper (ref 2) achieved the limit of detection of 10 aM without rHPCR technique, which seems better than the one in this work.
7) The authors emphasized that the use of RNA incorporated primer can suppress the non-specific amplification signal produced in the absence of target RNA, but undesirable amplification in the absence of RNA was observed in Figure 4. The authors need to discuss why these background signals do not completely disappear.
8) The authors need to check the specificity of the proposed system by using the control RNA samples.
9) Instead of listing the primer sequences, the authors need to make the table for clarity.

---

## Round 0.2 · accepted · Accept

Thanks for addressing all the revisions and corrections requested. Now your manuscript is accepted in PeerJ.

·

Basic reporting

The reviewed version of the manuscript is clearly presented. Authors answered each of the reviewers´ questions in a proper way. Title is now more appropriated and professional english was used throughout the manuscript. Figures are well presented, and a table was added to show the primers´ sequences utilized in experiments. The paper is now fluid and readable. The paper structure is appropriated for Peer J.

Experimental design

All questions raised by reviewers are answered and proper reasons are given at each question raised. A legend about number of replicates was added to Figure 4. I consider this part is well described in the manuscript in a clear way.

Validity of the findings

Points to review and correct were tackled and now it is in a good shape. Authors improved Results and Discussion sections taking into account points raised during the first revision. Conclusion is clearly stated.

Additional comments

The paper "RNase H-dependent amplification improves the accuracy of rolling circle amplification combined with loop-mediated isothermal amplification (RCA-LAMP)" presents experimental evidence of the improvement of accuracy of the RCA-LAMP technique. The work is clearly presented, and the figures are precise.